# Reactivation of the PI3K/mTOR Signaling Pathway Confers Resistance to the FGFR4 Inhibitor FGF401

**DOI:** 10.3390/ijms26199818

**Published:** 2025-10-09

**Authors:** Hung Huynh, Wai Har Ng

**Affiliations:** Laboratory of Molecular Endocrinology, National Cancer Centre Singapore, Singapore 168583, Singapore; nmsnwh@nccs.com.sg

**Keywords:** patient-derived xenograft (PDX), hepatocellular carcinoma (HCC), FGF19/FGFR4 inhibitor, mTOR inhibitor, drug resistance, preclinical models, antitumor synergy, Raf, mitogen-activated protein extracellular kinase (MEK), extracellular signal-regulated kinase (ERK), phosphatidylinositol 3-kinase (PI3K)e, PI3K/AKT/mTOR pathway

## Abstract

Hepatocellular carcinoma (HCC) is a deadly liver cancer characterized by dysregulated signaling and aberrant cell-cycle control. The FGFR4/FGF19 pathway is dysregulated in HCC and other cancers. Inhibitors targeting the FGF19/FGFR4 pathway, including the FGF19/FGFR4 inhibitor FGF401, have been investigated in HCC and other cancers; however, nearly all patients who initially respond eventually develop resistance shortly after starting therapy, highlighting the urgent need for new treatment strategies to overcome drug resistance. In the present study, we report that chronic treatment of the FGF19/FGFR4-expressing HCC25−0705A line with FGF401 led to acquired resistance. FGF401-resistant tumors exhibited upregulation of FGFRs and activation of the PI3K/AKT/mTOR/p70S6K pathway. Combination therapy with FGF401 and the mammalian target of rapamycin (mTOR) inhibitor everolimus (FGF401/everolimus) resulted in more complete tumor growth inhibition, delayed the onset of resistance, and prolonged overall survival (OS) in mice bearing orthotopic HCC tumors. The FGF401/everolimus combination effectively suppressed tumor cell proliferation; promoted apoptosis; reduced tumor hypoxia via blood vessel normalization; and downregulated key proteins involved in proliferation, survival, metastasis, and angiogenesis. These preclinical findings provide a strong rationale for clinical trials combining FGFR4 and mTOR inhibitors in HCC patients with FGF19/FGFR4/mTOR-dependent tumors.

## 1. Introduction

HCC ranks as the third leading cause of cancer-related deaths worldwide, with a high mortality rate and a <10% 5-year survival rate [1]. HCC arises from various etiologies, including cirrhosis caused by hepatitis B and/or C infection, excessive alcohol consumption, diabetes mellitus, and non-alcoholic fatty liver disease [2]. Many HCC cases are inoperable and carry a very poor prognosis [3], with limited therapeutic options available. Curative treatments such as local ablation, surgical resection, or liver transplantation are suitable for only a small subset of patients [4,5]. For patients often diagnosed with advanced or metastatic HCC, local tumor destruction, chemoembolization, and systemic therapy remain treatment options. The multireceptor tyrosine kinase inhibitors sorafenib [6,7] and lenvatinib [8] have been established as standard systemic treatments for patients with advanced HCC in three randomized trials. Although sorafenib and lenvatinib demonstrated survival benefits [6,7,8], their impact is modest and frequently accompanied by the development of drug resistance [6,7]. The presence of intratumoral heterogeneity and hepatic cancer stem cell (CSC) subpopulations, which contribute to local recurrence and therapeutic resistance, further complicates HCC management [9,10,11,12]. In the second-line setting, the multikinase inhibitors regorafenib [13] and cabozantinib [14] have been approved after patients with advanced HCC showed significantly improved survival compared to those on the placebo. Immunotherapy, particularly immune checkpoint inhibitors (ICIs) such as anti-PD-1 and anti-CTLA-4, has transformed the treatment landscape by enhancing antitumor immune responses [15]. Systemic therapies combining atezolizumab and bevacizumab or durvalumab and tremelimumab are now the preferred first-line option [2], except in patients with high-risk stigmata of varices, gastrointestinal bleeding or advanced liver cirrhosis [16]. Given the aggressive nature of HCC, novel strategies to improve current therapies and clinical outcomes remain urgently needed.

In recent years, metabolic reprogramming and targeted inhibition of key signaling pathways have emerged as potential strategies to overcome treatment resistance and enhance survival rates [17]. Fibroblast growth factor 4 (FGFR4) expression has been reported to be significantly higher in HCC samples [18,19] and is associated with an advanced TNM stage and a shorter OS compared to tumors without FGFR4 expression [20]. FGF19 overexpression and FGFR4 expression have been detected in 14% and approximately 50% of HCC patients, respectively [21,22], and are correlated with poorer prognosis, earlier recurrence, faster tumor progression, and a shorter OS [22,23]. Consistent with these findings, we recently observed FGF19 overexpression in 12.5% of HCC patient-derived xenografts (PDXs) [19]. FGF19 has high specificity for FGFR4, and upon binding to FGFR4/β-Klotho, FGFR4 [24] and its downstream pathways are activated, including Ras/Raf/ERK1/2 and PI3K/AKT/mTOR [25]. Aberrant activation of the FGF19/FGFR4 axis plays a critical role in promoting cancer cell proliferation, differentiation, survival, and migration [19,23,26]. Overexpression of FGF19 in transgenic mice induces liver tumors that are sensitive to treatment with FGFR4 or FGF19 antagonist antibodies [21,27,28]. Furthermore, genetic knockout of FGFR4 prevents tumor development in transgenic mice with exogenous FGF19 expression. Together, these findings support the pathogenic role of aberrant FGF19/FGFR4 signaling in HCC and provide a strong rationale for targeting this pathway as a therapeutic strategy in the subset of HCC patients with FGF19/FGFR4 activation [21,28,29,30].

FGF401 is a novel, orally administered FGFR4 kinase inhibitor currently under clinical investigation [19]. It has demonstrated strong antitumor activity in mice bearing HCC xenografts and PDX models positive for FGF19, FGFR4, and KLB [19,31,32]. A completed Phase I/II study of FGF401 in patients with HCC or other solid tumors expressing FGFR4 and KLB (NCT02325739) showed its clinical efficacy, with objective responses observed (one complete response and seven partial responses; four each in Phases I and II), along with its manageable safety profile [33]. However, despite initial responses, resistant colonies emerged after prolonged FGF401 treatment, consistent with previous reports [19]. Ongoing studies are evaluating FGF401 in combination with other agents to overcome this resistance [34].

Aberrant activation of the PI3K/AKT/mTOR pathway occurs in nearly 50% of HCC cases [35,36] and is associated with poorly differentiated tumors, poor prognosis, and earlier recurrence [37,38], as well as increased tumor invasion and metastasis [39]. Forced overexpression of mTOR promotes cellular proliferation, prolongs cell survival, and inhibits apoptosis [40]. Key downstream targets of mTOR include ribosomal p70S6 kinase (p70S6K) and the eukaryotic initiation factor 4E-binding proteins (4E-BPs). Phosphorylation of 4E-BPs by mTOR releases them from eIF4E, enabling cap-dependent translation initiation in cancer cells [41,42]. Consequently, the PI3K/AKT/mTOR pathway is recognized as a central regulator of protein synthesis, angiogenesis, metastasis, proliferation, differentiation, and survival. Thus, mTOR represents a prominent therapeutic target in HCC.

Our recent study demonstrated that the mTOR inhibitor everolimus effectively suppressed tumor growth and significantly increased both total and functional blood vessels while alleviating tumor hypoxia in mTOR-dependent HCC PDX models [43]. In contrast, everolimus has been shown to exert antiangiogenic effects in several mTOR-independent experimental tumor models [44,45,46,47]. A global phase III trial of everolimus for liver cancer (EVOLVE-1) in patients progressing on sorafenib failed to meet its primary endpoint of improving OS [48]. Beyond the cytostatic effects of mTOR inhibition [45], compensatory activation of alternative signaling pathways—arising from mTORC1-mediated suppression of negative feedback loops—may limit the efficacy of everolimus as a single agent anticancer therapy. Combination strategies are therefore anticipated to provide greater benefits. Indeed, sirolimus analogs have been shown to enhance the antineoplastic activity of conventional cytotoxic drugs such as doxorubicin or vinblastine [49,50,51]. Consistent with this, our recent studies demonstrated that everolimus combined with vinorelbine [43] or sirolimus combined with bevacizumab [36] inhibited HCC growth to a greater extent than either monotherapy.

Here, we report that long-term inhibition of the FGF19/FGF4 signaling pathway by FGF401 leads to acquired resistance through reactivation of the PI3/AKT/mTOR pathway. We show that dual inhibition of mTOR and FGF19/FGFR4 signaling not only achieves more complete tumor growth suppression in FGF19/FGFR4-dependent or mTOR-dependent models and the FGF401-resistant HCC25–0705A−FGF401−R3 model than either agent alone but also delays the onset of FGF401 resistance and prolongs OS in mice bearing orthotopic HCC tumors. This combination effectively inhibited tumor cell proliferation; induced apoptosis; alleviated tumor hypoxia via blood vessel normalization; and downregulated multiple critical signaling pathways involved in proliferation, survival, metastasis, and angiogenesis. Importantly, the combination was well tolerated and outperforms monotherapies in suppressing tumor growth across multiple HCC PDX models. Our findings provide a strong rationale for a novel combination therapy targeting FGF19/FGFR4 and mTOR pathways in HCC.

## 2. Results

### 2.1. Prolonged Treatment with FGF401 Resulted in the Emergence of FGF401 Resistance

FGF401 has shown encouraging antitumor activity in FGF19/FGFR4-dependent HCC models [19] and in patients with HCC or solid tumors expressing FGFR4 and KLB [33]. Although FGF401 demonstrated clinical efficacy with objective responses (one complete response and seven partial responses; four each in Phases I and II) and an acceptable safety profile, its overall impact was modest and frequently accompanied by the development of drug resistance [33]. To investigate the mechanisms underlying FGF401-induced resistance, resistant HCC models were generated by continuous dosing of mice bearing the HCC25–0705A model, which expresses a high level of FGF19 and is highly sensitive to FGF401 at the clinically relevant dose of 30 mg/kg twice daily. As shown in Figure 1A, FGF401 initially inhibited HCC25–0705A tumor growth, consistent with our previous report [19]. However, with prolonged treatment, tumors treated with FGF401 alone began to show resistance at approximately day 30, followed by exponential progression in the subsequent days. The progressed HCC25–0705A tumors were harvested for serial transplantation, and after three serial passages in mice under continuous FGF401 treatment, the resistant HCC25–0705A–FGF401–R3 model was established.

To identify compensatory signaling pathways activated in response to FGF401 in the resistant HCC25–0705A–FGF401–R3 model, we performed Western blot analyses on parental HCC25–0705A and resistant HCC25–0705A–FGF401–R3 tumors treated with FGF401 (30 mg/kg, twice daily) for 5 days. In parental HCC25–0705A tumors, the 125 kDa glycosylated FGFR4 and its downstream effectors—including p-p70S6K (Thr421/Ser424), p-4EBP1 (Thr70), p-ERK1/2, Bcl-x, Cdc25C, p-Cdc25C, Cdc2, p-Cdc2, p-Cdk2, and p-Rb—were significantly decreased, while the levels of non-glycosylated 95 kDa FGFR4, FGFR3, cleaved caspase-3, p-AKT (Ser473), p-p70S6K (Thr389), and p-S6K (Ser235/236) were significantly increased by FGF401 (as shown in Figure 1B). In contrast, in HCC25–0705A–FGF401–R3, the levels of p-p70S6K (Thr421/Ser424), p-S6K (Ser235/236), and p-p4EBP1 (Thr70) were not significantly altered by FGF401 treatment compared with untreated parental HCC25–0705A (Figure 1B).

Furthermore, FGFR2, FGFR3, non-glycosylated 95kDa FGFR4, p-PI3Kp85 (Tyr458)/p55 (Tyr199), p-p70S6K (Thr389), p-S6K (Ser235/236), p27, p-Cdc2 (Tyr15), p-Cdk2 (Thr14/Tyr15), and p-Rb (Ser780) were significantly elevated in four independent resistant clones (A–D), even in the presence of FGF401 (Figure 1B). p-ERK1/2, which was markedly reduced by FGF401 in parental HCC25–0705A, was undetectable in all resistant clones, suggesting that feedback reactivation occurred primarily through PI3K/mTOR signaling rather than the ERK1/2 pathway.

Despite FGF401 treatment, resistant clones showed significant upregulation of Bcl-x and Survivin, along with reduced cleaved caspase-3, compared with parental HCC25–0705A and vehicle controls. These findings suggest that resistance to FGF401 arises from suppression of FGF401-induced apoptosis, likely mediated by Survivin and Bcl-x upregulation (Figure 1B). Given that secondary activation of the PI3K/AKT/mTOR pathway is a well-recognized mechanism of resistance to FGFR and other RTK inhibitors [52], we hypothesize that mTOR blockade could disrupt this feedback loop and overcome FGF401 resistance. We therefore evaluated the efficacy of combined FGF401 and everolimus treatment in HCC PDX models and in the resistance HCC25–0705A–FGF401–R3 model in vivo.

### 2.2. Dose-Dependent Inhibition of FGF401/Everolimus Growth

A major clinical concern is that combining FGF401 and everolimus at their standard monotherapy doses may lead to severe adverse effects. To address this, we investigated whether lower doses of both agents could achieve comparable antitumor efficacy in our models. Mice bearing high FGF19-expressing HCC26–0808B xenografts were treated with the vehicle or three step-down dose combinations of FGF401 and everolimus (30:1.0, 20:0.75, and 15:0.5 mg/kg) for 13 days. Treatments resulted in modest increases in body weight (Appendix A Appendix A). The combinations reduced the tumor burden by approximately 98.4%, 96.1%, and 94.2%, respectively (*p* < 0.0001). No noticeable clinical signs of toxicity were observed in any of the treatment groups compared with vehicle controls, indicating good tolerability (Appendix A Appendix A). Among the three dose groups, 30 mg/kg FGF401 plus 1 mg/kg everolimus achieved the strongest efficacy while maintaining a similar safety profile to the lower-dose groups, suggesting that 30 mg/kg FGF401 plus 1 mg/kg everolimus provides optimal antitumor activity with acceptable toxicity (Appendix A Appendix A). Based on these findings, we next evaluated this combination regimen in selected HCC PDX models.

### 2.3. FGF401/Everolimus Combination Reverses Resistance to FGF401

Figure 2 shows the body weight, tumor volume, tumor images, and tumor weight for parental HCC25–0705A (Figure 2A–D) and resistant HCC25–0705A–FGF401–R3 (Figure 2E–H) xenografts treated with FGF401, everolimus, or the FGF401/everolimus combination. In parental HCC25–0705A (*n* = 8/group), the mean tumor volume on day 12 was 1550 ± 150 mm^3^ for vehicle-treated mice, 180 ± 23 mm^3^ for mice treated with FGF401 (30 mg/kg, twice daily), 635 ± 54 mm^3^ for those treated with everolimus (1 mg/kg, once daily), and 70 ± 12 mm^3^ for the FGF401/everolimus group (Figure 2B).

Figure 2F shows the tumor growth in the resistant HCC25–0705A–FGF401–R3 model. On day 22, the mean tumor volumes were 2209 ± 350 mm^3^ for the vehicle, 1960 ± 410 mm^3^ for FGF401, 1202 ± 22 mm^3^ for everolimus, and 370 ± 22 mm^3^ for FGF401/everolimus combination. Compared with the parental line, HCC25–0705A–FGF401–R3 tumors exhibited similar sensitivity to the mTOR inhibitor everolimus alone (Figure 2B,F).

In parental HCC25–0705A, the FGF401/everolimus combination produced a robust response, reducing the tumor volume by ~75–80% relative to day 1 (Figure 2B). In contrast, in the resistant HCC25–0705A–FGF401–R3 model, FGF401 monotherapy showed no significant activity (Figure 2F). Notably, the combination began to suppress tumor growth by day 7, with significant shrinkage observed by day 18 (Figure 2F).

Consistent with our previous study [19], FGF401 significantly inhibited tumor growth in HCC25–0705A tumors highly expressing FGF19/FGFR4 (Figure 2B). However, FGF401 monotherapy had no effect in the resistant HCC25–0705A–FGF401–R3 tumors (Figure 2F). The addition of everolimus to FGF401 significantly enhanced the antitumor efficacy compared with either agent alone in both the parental and resistant models (Figure 2B,F). A modest increase in body weight was observed in everolimus- and FGF401/everolimus-treated groups compared with vehicle controls (Figure 2A,E). No other significant clinical signs of toxicity were detected in any treatment group.

### 2.4. FGF401/Everolimus Exerts Long-Lasting Tumor Growth Inhibition in PDX Models Expressing FGF19/FGFR4 or Both FGF19/FGFR4 and Activated mTOR

We next examined the therapeutic effect of FGF401/everolimus in HCC PDX models expressing FGF19/FGFR4, such as HCC09–0913 and HCC29–1104, or both FGF19 and activated mTOR, such as HCC2–1318, HCC26–1004, and HCC26–0808B. Treatment was initiated once xenografts reached approximately 100–250 mm^3^. Mice were randomized into four groups and treated orally with (a) vehicle (200 µL), (b) FGF401 (30 mg/kg, twice daily), (c) everolimus (1 mg/kg, once daily), or (d) FGF401 (30 mg/kg, twice daily) plus everolimus (1 mg/kg, once daily). Tumor volumes of HCC26–1004 (Figure 3A), HCC2–1318 (Figure 3D), HCC26–0808B (Appendix A Appendix A), HCC29–1104 (Appendix A Appendix A), and HCC09–0913 (Appendix A Appendix A) are shown. FGF401 significantly inhibited tumor growth across these xenograft lines. Notably, in the HCC26–0808B model, tumors treated with FGF401 alone began to show resistance around day 12, followed by exponential progression (Appendix A Appendix A). Among these five models examined, everolimus was as potent as FGF401 in suppressing tumor growth in HCC26–1004, HCC2–1318, and HCC26–0808B, which are both mTOR- and FGF19/FGFR4-dependent (Figure 3A,D; Appendix A Appendix A). By contrast, in HCC09–0913, which is mTOR-independent but FGF19/FGFR4-dependent, everolimus alone had no effect, whereas FGF401 potently suppressed growth (Appendix A Appendix A).

Combination treatment with FGF401/everolimus exerted a potent antitumor effect, resulting in maximal suppression of tumor growth in all five models compared with the vehicle controls (Figure 3A,D; Appendix A Appendix A). Strikingly, the combination produced a prolonged antitumor effect: the volumes of HCC26–1004, HCC2–1318, HCC26–0808B, and HCC09–0913 tumors were reduced to approximately 3%, 3%, 9%, and 1% of that of vehicle-treated tumors, respectively (Figure 3A,D; Appendix A Appendix A). In this long-term treatment, inhibition of mTOR clearly prolonged the response to FGF401. These results were consistent with tumor weight at the end of the treatment cycle (Figure 3C,D; Appendix A Appendix A).

Furthermore, FGF401/everolimus was superior to standard sorafenib treatment in inhibiting tumor growth in HCC26–1004 (Figure 3B,C) and HCC2–1318 (Figure 3E,F). All treated mice maintained a healthy coat, normal food and water intake, normal social interactions, and no signs of aggression, indicating that the administered doses resulted in minimal toxicity (Appendix A Appendix A). Importantly, all PDX models treated with FGF401/everolimus, including those relatively resistant to everolimus monotherapy (e.g., HCC09–0913), exhibited T/C ratios <0.37, surpassing the 0.42 threshold defined by the Investigational Drug Branch (IDB), Division of Cancer Treatment, National Cancer Institute [53].

### 2.5. Effects of FGF401, Everolimus, and FGF401/Everolimus on Liver and Kidney Injury-Related Parameters

To further assess whether the administered doses resulted in minimal toxicity and side effects, we analyzed liver and kidney injury-related parameters in sera from mice bearing HCC25–0705A xenografts treated with vehicle, FGF401 (30 mg/kg, twice daily), everolimus (1 mg/kg, once daily), or the combination of FGF401 (30mg/kg, twice daily) plus everolimus (1 mg/kg, once daily) for 12 days. As shown in Table 1, daily treatment with everolimus led to modest elevations in alanine aminotransferase (ALT), alkaline phosphatase (ALP), aspartate aminotransferase (AST), and total bilirubin (TBIL). This observation is consistent with the safety profile of everolimus in human studies [48,54,55], where elevations of these enzymes occur in up to one-quarter of patients but are typically mild and rarely necessitate dose modification or discontinuation. Compared with the vehicle, FGF401 caused approximately 3.4-fold and 2-fold increases in ALT and AST, respectively, suggesting mild liver dysfunction. No significant increases in blood urea nitrogen (BUN), ALT, ALP, AST, or TBIL were observed with the FGF401/everolimus combination compared with FGF401 monotherapy. Similarly, no significant changes in serum glucose (GLU) or albumin (ALB) were detected across treatment groups relative to the vehicle. Collectively, these data suggest that the FGF401/everolimus combination induced only mild hepatic toxicity.

### 2.6. Everolimus Synergizes with FGF401 to Normalize Blood Vessels, Inhibit Cell Proliferation, and Promote Tumor Cell Apoptosis in HCC Models

We next performed immunohistochemistry (IHC) to evaluate apoptotic and proliferative cells in post-treatment tumors. In HCC25–0705A, FGF401 modestly reduced p-histone H3 (Ser10)-positive cells but significantly increased cleaved PARP-positive cells (Figure 4A; Appendix A Appendix A). The combined FGF401/everolimus treatment further reduced p-histone H3 (Ser10)-positive cells and increased cleaved PARP-positive cells compared with FGF401 monotherapy, indicating enhanced inhibition of proliferation and induction of apoptosis (Figure 4A; Appendix A Appendix A). A similar trend was observed in the mTOR-dependent or dual FGF19/mTOR-dependent HCC26–0808B models, where FGF401 modestly inhibited proliferation and induced apoptosis (Appendix A Appendix A). In high-FGF19-expression PDX models, including HCC09–0913 (Appendix A Appendix A), HCC10–0112B (Appendix A Appendix A), and HCC01–0207 (Appendix A Appendix A), FGF401 markedly suppressed proliferation and promoted apoptosis. In HCC29–1104, FGF401 monotherapy modestly decreased p-histone H3 (Ser10)-positive cells and increased cleaved PARP-positive cells (Appendix A Appendix A). Across all six PDX models, the FGF401/everolimus combination consistently produced the lowest proportion of p-histone H3 (Ser10)-positive cells and the highest proportion of cleaved PARP-positive cells compared with either monotherapy group (Figure 4A; Appendix A Appendix A). These findings suggest that FGF401 and everolimus exert complementary effects to suppress proliferation and induce apoptosis in HCC models.

In HCC25–0705A, everolimus caused only a modest increase in total blood vessel counts (Figure 4A; Appendix A Appendix A), whereas no significant changes were observed in HCC09–0913 (Appendix A Appendix A), HCC10–0112B (Appendix A Appendix A), HCC29–1104 (Appendix A Appendix A), or HCC01–0207 (Appendix A Appendix A) compared with vehicle controls. These findings indicate that everolimus has weak or negligible antiangiogenic activity in these models. By contrast, FGF401 and FGF401/everolimus treatments significantly increased blood vessel density in HCC09–0913, HCC29–1104, HCC01–0207, HCC25–0705A, HCC26–0808B, and HCC10–0112B PDX models (Figure 4A; Appendix A Appendix A). Importantly, many of these vessels were functional, as indicated by lectin-positive staining in FGF401-treated HCC25–0705A (Figure 4A; Appendix A Appendix A), everolimus-treated HCC25–0705A (Figure 4A; Appendix A Appendix A), HCC26–0808B (Appendix A Appendix A), and HCC29–1104 tumors (Appendix A Appendix A). In contrast, vehicle-treated tumors displayed minimal or no lectin staining, suggesting non-functional vasculature.

We next examined tumor hypoxia using pimonidazole (PIMO) staining. In HCC25–0705A (Figure 4), HCC26–0808B (Appendix A Appendix A), and HCC29–1104 (Appendix A Appendix A), FGF401, everolimus, or their combination markedly reduced hypoxyprobe staining, indicating well-oxygenated tumor regions. Notably, in both FGF401- and everolimus-sensitive models, the combination of FGF401/everolimus did not significantly alter the total blood vessel density, lectin-positive vessel counts, or hypoxia compared with monotherapy. Collectively, these data suggest that everolimus does not antagonize the effects of FGF401 on vessel density, vascular normalization, or tumor oxygenation.

In contrast to parental HCC25–0705A, sections from the resistant HCC25–0705A–FGF401–R3 tumors showed only modest changes in p-histone H3 (Ser10)-positive and cleaved PARP-positive cell percentages following FGF401 or everolimus treatment (Figure 4B; Appendix A Appendix A). However, the combination of FGF401 with everolimus significantly enhanced antiproliferation and pro-apoptotic effects, as indicated by a further reduction in p-histone H3 (Ser10)-positive cells and an increase in cleaved PARP-positive cells (Figure 4B; Appendix A Appendix A).

Unlike the parental model, HCC25–0705A–FGF401–R3 tumors no longer displayed FGF401-induced increases in total blood vessel density or vascular normalization (Figure 4B). Strikingly, the addition of everolimus restored these effects, leading to ~2.5- and ~4.5-fold more lectin-positive blood vessels than everolimus or FGF401 monotherapy, respectively (Figure 4B; Appendix A Appendix A). Micrographs of vehicle-treated HCC25–0705A–FGF401–R3 tumors revealed hypoxic regions interspersed with vasculature, suggesting poor oxygen delivery despite vessel presence (Figure 4B; Appendix A Appendix A). Both FGF401 and everolimus monotherapy reduced intratumoral hypoxia compared with the vehicle, whereas the combination treatment eliminated hypoxyprobe staining entirely, indicating well-oxygenated tumor tissue (Figure 4B). Taken together, the perfusion and hypoxia analyses demonstrate that dual inhibition of FGF19/FGFR4 and mTOR signaling promotes the formation of a functional vascular network in high-FGF19-expression tumors. Beyond suppressing tumor growth, FGF401 and FGF401/everolimus reduce tumor hypoxia by generating normalized, well-perfused, capillary-like vessels. These findings suggest that mTOR contributes to maximal growth inhibition in part by promoting vascular normalization, alleviating hypoxia, inhibiting proliferation, and enhancing apoptosis in FGF19/FGFR4-dependent tumors.

### 2.7. Everolimus Abrogated FGF401-Induced Activation of mTOR but Not MEK/ERK Pathways

To elucidate the mechanisms underlying the potent antitumor activity of the FGF401/everolimus combination, we treated HCC26–0808B, HCC10–0112B, and HCC01–0207 tumors with FGF401, everolimus, or their combination. Western blot analysis revealed elevated levels of FGFR1, FGFR3, and 125 kDa FGFR4 in HCC26–0808B (Figure 5), as well as increased FGFR3 and 125 kDa FGFR4 in HCC10–0112B (Appendix A Appendix A) after everolimus monotherapy. A modest increase in FGFR1, FGFR3, and FGFR4 was also observed in everolimus-treated HCC01–0207 tumors (Appendix A Appendix A). FGF401 monotherapy modestly reduced FGFR1 and both 125 kDa/95 kDa FGFR4 levels in HCC26–0808B but increased 95 kDa FGFR4 levels in HCC01–0207 and HCC10–0112B tumors. Notably, the 95 kDa FGFR4 isoform exhibited faster migration than that in vehicle-treated lysates, likely due to the reduced glycosylation observed across all three PDX models (Figure 5; Appendix A Appendix A).

Because proliferation is primarily mediated by ERK signaling, whereas cancer cell survival is regulated by PI3K/AKT/mTOR signaling, we next examined ERK1/2 and mTOR activation in post-treatment HCC26–0808B, HCC10–0112B, and HCC01–0207 tumors. Western blot analysis revealed that p-ERK1/2 levels in HCC26–0808B tumors were unaffected by FGF401 treatment (Figure 5). In contrast, p-ERK1/2 was barely detectable in FGF401-treated HCC10–0112B (Appendix A Appendix A) but was significantly increased in HCC01–0207 (Appendix A Appendix A). Adding everolimus to FGF401 did not further alter the p-ERK1/2 levels in HCC26–0808B or HCC10–0112B compared with FGF401 alone (Figure 5; Appendix A Appendix A), but markedly increased p-ERK1/2 levels in HCC01–0207 tumors (Appendix A Appendix A).

As shown in Figure 5, the levels of p-p70S6K (Thr421/Ser424), p-4EBP1 (Thr70), and p-S6R (Ser235/236) were significantly decreased in everolimus- and FGF401/everolimus-treated HCC26–0808B tumors. Consistent with the inhibitory role of FGFR4/FGF19 on mTOR signaling and cell-cycle progression, FGF401 treatment also suppressed p-p70S6K (Thr421/Ser424), p-4EBP1 (Thr70), and p-S6R (Ser235/236), as well as Cdc25C, p-Cdk2 (Thr14/Tyr15), Survivin, Cdc2, and p-Rb (Ser780) (Figure 5). Compared with FGF401 monotherapy, the combination treatment of FGF401/everolimus led to further reductions in 125 kDa FGFR4, p-p70S6K (Thr421/Ser424), p-4EBP1 (Thr70), p-S6R (Ser235/236), and p-Cdc2 (Tyr15). While FGF401 monotherapy increased cleaved caspase-7 levels, the combination therapy led to the highest cleaved caspase-7 expression compared with the vehicle and monotherapies (Figure 5).

In HCC10–0112B, everolimus significantly increased the levels of p-AKT (Ser473), p-p70S6K (Thr421/Ser424), Survivin, p-Cdc25C (Ser216), and Cdc25C, suggesting the activation of feedback loop mechanisms (Appendix A Appendix A). Notably, these feedback effects were abolished by FGF401/everolimus treatment, which concurrently elevated cleaved caspase-7 expression, indicating enhanced apoptosis (Appendix A Appendix A). In HCC01–0207, the combination of FGF401/everolimus was more effective than either monotherapy in reducing the levels of p-p70S6K (Thr421/Ser424), p-S6R (Ser235/236), p-4EBP1 (Thr70), Cyclin D1, p-Cdk2 (Thr14/Tyr15), p-Cdc2 (Tyr15), Survivin, p-Rb (Ser780), and Cdc25C. Consistently, FGF401/everolimus also induced greater apoptosis, as evidenced by elevated cleaved caspase-7 levels (Appendix A Appendix A).

### 2.8. FGF401/Everolimus Prolongs the Survival of Mice Bearing HCC Orthotopic Tumors

Given that the local microenvironment is critical for tumor growth, we investigated whether everolimus-induced suppression of the mTOR pathway could overcome resistance to FGF401, thereby inhibiting tumor growth and improving survival in mice bearing HCC tumors. We evaluated the effects of FGF401/everolimus in orthotopic HCC09–0913, HCC25–0705A, and HCC25–0705A–FGF401–R3 models. Mice with established tumors received treatment with FGF401 (30 mg/kg, twice daily), everolimus (1 mg/kg, once daily), or their combination for 28 days. Kaplan–Meier survival analysis showed that all mice treated with FGF401 or everolimus alone became moribund by day 135 and day 85, respectively, in HCC09–0913; by day 72 and day 130, respectively, in parental HCC25–0705A; and by day 56 and day 88, respectively, in HCC25–0705A–FGF401–R3 (Figure 6). In contrast, the FGF401/everolimus combination group exhibited significantly prolonged survival, with moribund endpoints at day 180 in HCC09–0913 and HCC25–0705A and day 144 in HCC25–0705A–FGF401–R3 (Figure 6; *p* < 0.01, log-rank test). Notably, no significant differences in body weight were observed among treatment groups. Together, these findings demonstrate that the FGF401/everolimus combination is superior to either monotherapy in improving OS in mice bearing orthotopic HCC tumors.

## 3. Discussion

HCC ranks as the seventh most commonly diagnosed cancer worldwide and is the third leading cause of cancer-related mortality [1,56]. Despite recent therapeutic advances, the treatment options remain limited, and improvements in OS have been only modest [56]. FGF19 overexpression and FGFR4 expression are detected in approximately 14% and 50% of HCC patients, respectively [21,22], and are associated with poorer prognosis, higher recurrence rates, faster tumor progression, and shorter OS [22,23]. In our recent studies, FGF19 overexpression was observed in 12.5% of HCC PDX models [19], and the FGF19/FGFR4 pathway was shown to play a key role in promoting the proliferation, differentiation, apoptosis, and migration of cancer cells [19]. Inhibitors targeting the FGF19/FGFR4 and PI3K/mTOR/ p70S6K pathways have been investigated across various cancers, including HCC. Notably, FGF401 effectively suppressed growth in FGF19/FGFR4-positive HCC models [19] and has been clinically evaluated in the phase I/II trial NCT02325739 in patients with HCC or other solid tumors expressing FGFR4 and KLB [33]. This study demonstrated manageable safety and tolerability, with the most common treatment-related adverse events being diarrhea, nausea, and elevated liver enzymes. In the HCC cohort, preliminary antitumor activity was observed, including partial responses and disease stabilization in patients with FGF19-positive tumors. However, despite the initial response, we observed the emergence of resistant colonies following prolonged FGF401 treatment, consistent with previous reports [19].

In this study, we used the FGF19/FGFR4-dependent HCC25–0705A model to generate an acquired FGF401-resistant derivative, HCC25–0705A–FGF401–R3 (Figure 1). Although HCC25–0705A tumors initially respond to FGF401 with shrinkage, complete regression is rare. With continued treatment, a subpopulation of quiescent, FGF401-tolerant cells eventually resume proliferation, giving rise to a resistant population (Figure 1). Gatekeeper (V550M/L) and hinge-1 (C552) mutations in the FGFR4 kinase domain have been reported in HCC tumors following treatment with the FGFR4 inhibitor fisogatinib (BLU-554) [57]; however, no FGFR4 mutations were detected in HCC25–0705A–FGF401–R3 tumors. To investigate the mechanisms underlying resistance, we performed Western blot analyses of parental HCC25–0705A and resistant HCC25–0705A–FGF401–R3 tumors treated with clinically relevant concentrations of FGF401. The aim of this approach was to uncover feedback signaling circuits and nodes that could be targeted by synergistic drug combinations. Western blot analysis revealed upregulation of multiple signaling pathways in HCC25–0705A–FGF401–R3 tumors despite FGF401 exposure. In resistant tumors, phosphorylation of PI3K p85 (Tyr458)/p55 (Tyr199) and p70S6K (Thr389) was elevated under acute FGF401 treatment. Moreover, p-p70S6K (Thr421/Ser424), p-S6R (Ser235/236), and p-4EBP1 (Thr70) levels were not significantly reduced by FGF401, in contrast to the marked inhibition observed in parental HCC25–0705A tumors (Figure 1B). Notably, phosphorylation of S6R was further elevated, while p-ERK1/2 was abolished in HCC25–0705A–FGF401–R3, suggesting that mTOR pathway reactivation contributes to resistance through an ERK–independent mechanism. In addition, FGFR2 and FGFR3 were markedly upregulated in resistant tumors, potentially driving PI3K/AKT/mTOR activation. Apoptotic signaling was also impaired, as evidenced by increased Survivin and Bcl-x levels and reduced Bim and cleaved caspase-3 levels in HCC25–0705A–FGF401–R3 tumors. These findings align with prior studies demonstrating that sustained mTOR signaling is sufficient to maintain proliferation and survival, thereby promoting resistance to FGFR inhibitors such as PD173074 and AZD4547 in endometrial cancer [58] and AZD4547 in diffuse-type gastric cancer [59]. Collectively, these preclinical data provide a strong rationale for combining FGFR4 and mTOR inhibitors in FGF19/FGFR4/mTOR-dependent tumors.

The combination of FGF19/FGFR4 and mTOR inhibition with FGF401/everolimus not only achieved more complete tumor growth suppression in both FGF19/FGFR4-positive and FGF401-resistant HCC25–0705A–FGF401–R3 models than either monotherapy but also delayed the onset of FGF401 resistance and significantly prolonged OS in mice bearing orthotopic HCC tumors. Mechanistically, the combination effectively suppressed tumor cell proliferation, induced apoptosis, reduced hypoxia through vascular normalization, and downregulated multiple signaling pathways involved in proliferation, survival, metastasis, and angiogenesis. The response rates (T/C < 0.42) were 46.1% (6/13) for FGF401, 69.2% (9/13) for everolimus, and 100% (13/13) for the FGF401/everolimus combination. While all tumors treated with either monotherapy eventually developed resistance and exhibited numerous drug-resistant colonies, as evidenced by hematoxylin and eosin (H&E) staining, none of the tumors treated with FGF401/everolimus showed resistance, and only 1 of 10 (10%) harbored resistant colonies. Consistently, Western blot analysis confirmed that FGF401/everolimus effectively suppressed the critical signaling pathways driving proliferation, survival, metastasis, and angiogenesis, including cell-cycle regulators, the Raf/MEK/ERK pathway, and the PI3K/AKT/mTOR cascade. Taken together, these findings suggest that the effect of combining FGF401 with everolimus is more than additive and can be considered synergistic, as everolimus not only enhanced antitumor activity but also delayed the development of resistance to FGF401. This synergistic interaction likely results from the dual blockade of complementary oncogenic signaling pathways, consistent with prior studies showing enhanced efficacy when FGFR4 and PI3K/mTOR pathways are co-inhibited.

We found that everolimus abolished FGF401 resistance in the HCC25–0705A–FGF401–R3 model not only by further suppressing mTOR targets but also by inhibiting the parallel MEK/ERK pathway. This sensitization was mediated through the induction of apoptosis, inhibition of cell proliferation, promotion of vascular normalization, and reduction in tumor hypoxia. The FGF401/everolimus combination was highly efficacious in HCC tumors with FGF19 overexpression and/or PI3K/AKT/mTOR pathway activation, which occur in ~12.5% and ~50% of patients, respectively [19,43,60]. Consistently, it demonstrated robust antitumor activity across multiple FGF19/FGFR4-expressing PDX models (e.g., HCC09–0913 and HCC29–1104) and in tumors with both FGF19 overexpression and mTOR activation (e.g., HCC01–0207, HCC2–1318, HCC26–1004, HCC25–0705A, HCC26–0808B, and HCC10–0112B). Importantly, the combination also suppressed tumor growth and prolonged survival in orthotopic HCC PDX models. Collectively, these findings highlight dysregulation of the PI3K/mTOR signaling pathway as a key mechanism driving FGF401 resistance in HCC.

In recent years, combination therapies or dual-target inhibitors have been used clinically to improve drug delivery through blood vessel normalization. As shown in Figure 4, blood vessels in vehicle-treated HCC25–0705A tumors were large and dilated, with further dilation observed in vehicle-treated HCC25–0705A–FGF401–R3 tumors. IHC confirmed that FGF401 monotherapy induced blood vessel normalization and suppressed tumor cell proliferation in HCC25–0705A but not in HCC25–0705A–FGF401–R3. The combination treatment was even more effective, as evidenced by increased the CD31/lectin-positive vessel density and reduced tumor hypoxia. Loss of vessel normalization in HCC25–0705A–FGF401–R3 likely limits oxygen and drug delivery, necessitating higher doses that can lead to non-specific targeting and intolerable toxicity. The ability of FGF401/everolimus to enhance the antitumor effect of FGF401, overcome resistance, and remodel the tumor microenvironment through vascular normalization suggests its potential to address tumor heterogeneity by concurrently targeting multiple aberrant signaling pathways, including FGF19/FGFR4 and mTOR [61]. Such combination strategies may achieve synergistic effects and overcome the resistance mechanisms commonly observed in monotherapy.

Activation of the mTOR pathway has been associated with poorly differentiated tumors, a poor prognosis, and early recurrence in HCC [37,38], making it an attractive therapeutic target [62,63]. However, clinical trials targeting mTOR have largely failed to meet expectations [64,65,66,67,68]. In this study, we consistently observed FGFR3 upregulation in HCC25–0705A–FGF401–R3, HCC26–0808B, and HCC10–0112B models Figure 1B and Figure 5; Appendix A Appendix A), suggesting that activation of FGFR2 or FGFR3 may reduce FGF401 sensitivity by stimulating downstream mTOR targets, including p-p70S6K (Thr421/Ser424), p-4EBP1 (Thr70), p-S6R (Ser235/236), and Survivin, without affecting p-ERK1/2 in HCC25–0705A–FGF401–R3. Previous studies have shown that everolimus monotherapy can inhibit tumor growth and induce vascular normalization in mTOR-dependent models, but its capacity to trigger tumor cell death is limited [43,69]. Consistent with earlier findings [70,71], everolimus treatment induced AKT upregulation, likely by relieving S6K1-mediated inhibition of insulin receptor substrates, thereby activating PI3K/AKT signaling. Elevated Survivin and Bcl-x levels further attenuated the apoptotic activity of FGF401, as evidenced by reduced cleaved caspase-3 in FGF401-treated HCC25–0705A–FGF401–R3 tumors (Figure 1B). While everolimus alone modestly decreased p-4EBP1 (Thr70), p-S6R (Ser235/236), and Survivin, the FGF401/everolimus combination produced significantly greater reductions in these proteins, highlighting synergistic inhibitory mechanisms. Enhanced suppression of FGFRs and positive cell-cycle regulators likely contributes to the potent antitumor activity observed with the combination therapy.

In this study, we elucidated the mechanisms underlying FGF401 resistance in HCC PDX models and provide insights to overcome it. In the HCC25–0705A–FGF401–R3 model, resistance was linked to FGFR upregulation, enhanced PI3K phosphorylation, and sustained activation of downstream effectors, including p-p70S6K (Thr421/Ser424), p-4EBP1 (Thr70), and p-S6R (Ser235/236). Importantly, the combination of FGF401 with everolimus effectively inhibited HCC tumor growth driven by FGF19/FGFR4 or FGF19/FGFR4/mTOR signaling. This regimen not only delayed or prevented resistance in xenograft models but also markedly suppressed tumor progression in FGF401-resistant HCC25–0705A–FGF401–R3 tumors, where FGF401 monotherapy was ineffective. The enhanced efficacy was associated with reduced proliferation, increased apoptosis, and alleviated hypoxia via vascular normalization. Collectively, these findings highlight FGF401/everolimus as a promising therapeutic strategy for HCC with FGF19/FGFR4 activation, with or without concurrent mTOR/P70S6K signaling, warranting further evaluation in adjuvant or first-line metastatic settings. Given the diverse resistance mechanisms observed, molecular analyses of repeat biopsy specimens at the time of acquired resistance could provide comprehensive profiles of receptor tyrosine kinase expression and other pathways mediating FGF401 resistance in FGFR4/FGF19-driven HCC.

Although everolimus and sirolimus are widely used immunosuppressants and may alter immune homeostasis [72], the FGF401/everolimus combination demonstrated robust antitumor activity, suppression of metastasis, and manageable safety in preclinical studies [73]. Future investigations should assess whether reduced dosing of everolimus or sirolimus, combined with lower doses of FGF401, can mitigate immunosuppressive effects while preserving—or even enhancing—antitumor efficacy.

Overall, the findings from this study have important translational implications. In clinical practice, the combination of FGF401 with everolimus could offer a novel therapeutic strategy for patients with FGF19/FGFR4-positive HCC, particularly those who develop resistance to FGFR4 inhibitors. From a drug development perspective, these results provide a strong rationale for clinical trials evaluating dual inhibition of FGFR4 and mTOR pathways, with careful optimization of dosing schedules to balance efficacy and immune-related risks. Mechanistically, FGFR4 blockade mitigates feedback activation of the mTOR pathway, while mTOR inhibition counteracts compensatory FGFR signaling, thereby preventing the emergence of resistant clones. Taken together, our findings support evaluation of this combination in frontline adjuvant or first-line metastatic settings and highlight opportunities for integration with immunotherapy to further improve clinical outcomes.

## 4. Materials and Methods

### 4.1. Reagents

Everolimus (Selleck Chemicals LLC, Houston, TX, USA) was suspended in a vehicle solution containing 7 parts 30% (*w*/*v*) Captisol^®^ (sulfobutylether-β-cyclodextrin; CyDex Pharmaceuticals, Inc., Lenexa, KS, USA) and 3 parts PEG300, while FGF401 was dissolved in 100 mM citrate buffer (pH 2.5).

All primary antibodies used in the Western blot analyses are listed in Appendix A.

### 4.2. HCC Patient-Derived Xenograft (PDX) Models

This study was approved by the SingHealth Centralised Institutional Review Board (ethics code: CIRB #2006/435/B; approval date: 2 October 2018). All animal experiments were conducted in accordance with the Guide for the Care and Use of Laboratory Animals, published by the National Institutes of Health, USA [53]. Mice were housed in negative-pressure isolators under controlled conditions (23 °C, 43% humidity, 12 h light/dark cycles) with sterilized food and water provided ad libitum. All procedures were reviewed and approved by the Institutional Animal Care and Use Committee (IACUC).

In this study, the following HCC tumors were implanted into male C.B-17 severe combined immunodeficiency (SCID) mice (9–10 weeks old, 23–25 g; InVivos Pte. Ltd., Singapore) to establish PDX models, as described previously [19,74,75]: HCC01–0207, HCC2–1318, HCC09–0913, HCC10–0112B, HCC25–0705A, HCC26–0808B, HCC26–1004, and HCC29–1104. The FGF401-resistant HCC25–0705A–FGF401–R3 model was derived from the FGF19-expressing HCC25–0705A PDX model.

For dose titration studies, mice bearing high-FGF19-expression HCC26–0808B and HCC29–1104 xenografts (*n* = 8–10/group) were treated with the vehicle, everolimus, FGF401, or their combinations in a step-down dosing scheme (30:1, 20:0.75, and 15:0.5 mg/kg FGF401/everolimus) for the indicated duration. FGF401 was administered orally twice daily at a low dose of 30 mg/kg, whereas everolimus was administered once daily, as described previously [19].

### 4.3. Development of the FGF401-Resistant HCC25–0705A–FGF401–R3 Model

Mice bearing HCC25–0705A tumors were treated chronically with FGF401 (30 mg/kg, twice daily) for 40 days. Treatment was initiated when tumors reached ~150 mm^3^. After an initial response, HCC25–0705A tumors gradually developed resistance, leading to renewed tumor growth. Resistant tumors were harvested for serial transplantation when they reached ~1500 mm^3^. Mice bearing transplanted resistant tumors were re-treated with FGF401, and tumors were again harvested for implantation. This cycle was repeated until FGF401 had minimal impact on tumor growth.

FGF401 resistance was confirmed by the absence of significant tumor growth inhibition after 20–22 days of treatment. The third passage of resistant tumors (HCC25–0705A–FGF401–R3) was then established and subsequently used to evaluate the in vivo efficacy of the FGF401/everolimus combination.

### 4.4. Efficacy of FGF401/Everolimus in Ectopic HCC Models

FGF401 and everolimus were used to block FGF19/FGFR4 signaling and mTOR activity, respectively. In the combination experiments, tumor-bearing mice were randomized into four treatment groups (*n* = 8–10/group) and treated orally as follows: (a) vehicle control (100 mM citrate buffer, pH 2.5, with a mixture of 7 parts 30% (*w*/*v*) Captisol^®^ to 3 parts PEG300), (b) FGF401 (30 mg/kg, twice daily), (c) everolimus (1 mg/kg; once daily), and (d) FGF401 (30 mg/kg, twice daily) plus everolimus (1 mg/kg; once daily). Treatment was initiated when tumors reached approximately 100–250 mm^3^. Tumor growth was monitored, and tumor volume was calculated as described previously [74,75]. Mice were euthanized if they met any of the following humane endpoint criteria: >10% body weight loss, abdominal distension, abnormal posture or breathing, ruffled fur, impaired mobility, loss of appetite (reduced eating, drinking, or urination), or lack of interaction with cage mates. At study completion, mice were euthanized by CO_2_ asphyxiation 2 h after the final treatment. Body and tumor weights were recorded, and tumors were harvested for further analyses.

The efficacy of FGF401, everolimus, and their combination was assessed using the T/C ratio, where T and C represent the median tumor weights of drug-treated and vehicle-treated groups, respectively, at the end of treatment. A T/C ratio < 0.42 was considered indicative of antitumor activity [75]. All results were confirmed in independent experiments and are presented as mean ± SE.

### 4.5. Efficacy of FGF401, Everolimus, and FGF401/Everolimus in Orthotopic HCC Models

HCC09–0913, HCC25–0705A, and HCC25–0705A–FGF401–R3 orthotopic models were generated as previously described [75]. For the survival study, mice bearing tumors were randomized into four treatment groups (*n* = 10) and orally treated as follows: (a) vehicle control (100 mM citrate buffer, pH 2.5, with a mixture of 7 parts 30% (*w/v*) Captisol^®^ to 3 parts PEG300), (b) FGF401 (30 mg/kg, twice daily), (c) everolimus (1 mg/kg; once daily), and (d) FGF401 (30 mg/kg, twice daily) plus everolimus (1 mg/kg; once daily). Treatments were administered for 28 days, starting when tumor volumes reached approximately 100–150 mm^3^. Overall health status, body weight, and OS were monitored daily. Mice were euthanized upon meeting any of the following criteria: >10% body weight loss, abdominal distension, abnormal posture or breathing, ruffled fur, impaired mobility, loss of appetite (reduced eating, drinking, or urination), or lack of interaction with cage mates.

### 4.6. Western Blot Analysis and Quantification Analysis

To assess changes in protein expressions between vehicle- and drug-treated tumors, samples (*n* = 8–10/group) were homogenized in lysis buffer containing 50 mM Tris-HCl (pH 7.4), 150 mM NaCl, 0.5% NP-40, 1 mM EDTA, 10 mM Na_3_VO_4_, and 25 mM NaF, supplemented with protease inhibitors. Approximately 80 µg of protein per sample was resolved via sodium dodecyl sulfate–polyacrylamide gel electrophoresis (SDS–PAGE) and transferred to nitrocellulose membranes, as described previously [75]. Membranes were incubated with the indicated primary antibodies, followed by horseradish peroxidase-conjugated secondary antibodies. Protein bands were visualized with the WesternBright ECL HRP substrate (Advansta, Inc., San Jose, CA, USA) and detected on autoradiography film (Agfa Healthcare, Mortsel, Belgium). Films were scanned using a GS-900 Calibrated Densitometer (BioRad, Hercules, CA, USA).

For quantification analysis, band intensities were measured using Image Lab^TM^ software (Version 6.1; BioRad, Hercules, CA, USA), normalized to α-tubulin (loading control), and expressed as the fold change relative to the vehicle group. Values greater than 1 indicated increased expression, whereas values less than 1 indicated decreased expression compared with the control group.

### 4.7. Serum Analysis

Serum samples were collected from mice (*n* = 4/group) treated with vehicle, FGF401, everolimus, or FGF401/everolimus combination for 14 days. Biochemical parameters, including total bilirubin (TBIL), alkaline phosphatase (ALP), alanine aminotransferase (ALT), aspartate aminotransferase (AST), albumin (ALB), creatinine (CRE), glucose (GLU), and blood urea nitrogen (BUN), were measured using the VETSCAN^®^ Preventive Care Profile Plus (Abaxis Inc., Union City, CA, USA) according to the manufacturer’s instructions. Results are expressed as mean ± SE. Statistical significance was determined by one-way analysis of variance (ANOVA) followed by Tukey’s test (* *p* < 0.05).

### 4.8. Immunohistochemistry (IHC)

Tumor tissues were fixed in 10% buffered formalin (ICM Pharma Pte. Ltd., Singapore) at room temperature for 24 h and processed through graded ethanol, xylene, and paraffin using a Leica TP1020 Automatic Benchtop Tissue Processor (Leica Biosystems Nussloch GmbH, Nussloch, Germany). Sections (5 µm) were immunostained with antibodies against p-histone H3 (Ser10) (#9701, Cell Signaling Technology, Beverly, MA, USA), cleaved PARP (#5625, Cell Signaling Technology, Beverly, MA, USA), and CD31 (#77699, Cell Signaling Technology, Beverly, MA, USA) to evaluate cell proliferation, apoptosis, and blood vessel density, respectively, as described previously [75]. Slides were counterstained with hematoxylin (Sigma Diagnostics, St. Louis, MO, USA), dehydrated, and mounted. For quantification, at least 10 random fields per slide were captured at 100× magnification using an Olympus BX60 microscope (Olympus, Tokyo, Japan). Positively stained cells or vessels were counted in each field, and the results are expressed as mean ± SE.

### 4.9. Vessel Perfusion and Hypoxia Studies

Vessel perfusion and hypoxia studies were performed as previously described [75]. Mice bearing vehicle- or drug-treated tumors were intravenously injected with 100 µg of biotinylated tomato lectin (derived from *Lycopersicon esculentum*; Vector Labs, Newark, CA, USA, #B-1175) prepared in 100 μL of 0.9% NaCl. Tumors were harvested 10 min after injection, fixed in 10% buffered formalin, processed, and paraffin-embedded for sectioning. To visualize functional blood vessels, sections (5 µm) were stained using the streptavidin–biotin peroxidase complex method (Lab Vision Corporation, Fremont, CA, USA) according to the manufacturer’s instructions. The functional vessel density was quantified by counting lectin-positive vessels in 10 random 0.159 mm^2^ fields per tumor at 100× magnification.

For hypoxia analysis, mice were intraperitoneally injected with 60 mg/kg of pimonidazole hydrochloride (Hypoxyprobe Inc., Burlington, MA, USA) 1 h prior to tumor harvesting. Hypoxic regions were identified by IHC staining with the Hypoxyprobe Plus Kit HP2 (Hypoxyprobe Inc., Burlington, MA, USA) according to the manufacturer’s instructions. Quantification was performed by analyzing 10 random 0.159 mm^2^ fields per tumor at 100× magnification.

### 4.10. Statistical Analysis

Sample sizes were determined based on historical experimental data and power calculations (85% power). Statistical significance was defined as *p* < 0.05 (α = 0.05). Significance levels are indicated as follows: * *p* < 0.05; ** *p* < 0.01; *** *p* < 0.001.

Statistical analyses were performed using SigmaStat (Version 3.0, San Jose, CA, USA) and GraphPad Prism (Version 6.0, La Jolla, CA, USA). In vivo results were screened for outliers using Grubb’s test (http://www.graphpad.com accessed on 6 November 2023). For comparisons among multiple groups, one-way analysis of variance (ANOVA) followed by the Tukey–Kramer post hoc was applied.

Differences in protein expression levels, tumor volume, tumor weight, body weight at sacrifice, p-histone H3 (Ser10) index, cleaved PARP-positive cells, CD31-positive vessels, and lectin-positive vessels were analyzed. Error bars represent the standard deviation (SD). Survival analysis was performed using the log-rank test.

## Figures and Tables

**Figure 1 ijms-26-09818-f001:**
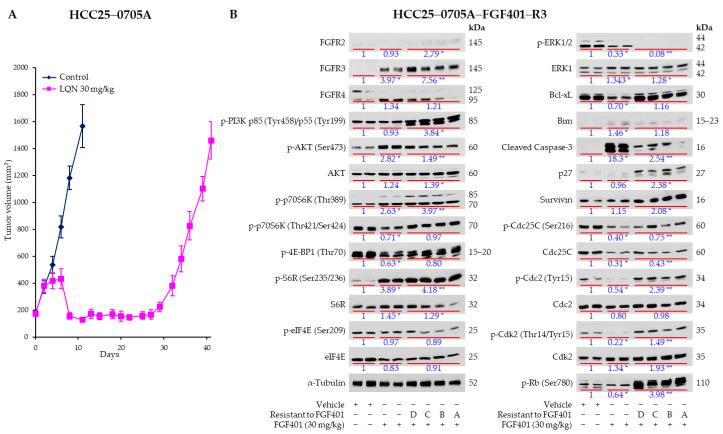
(**A**) Establishment of the FGF401-resistant HCC25–0705A (HCC25–0705A–FGF401–R3) model. Mice bearing HCC25–0705A xenografts expressing FGF19 were treated with the vehicle or 30 mg/kg FGF401 twice daily for 40 days, starting when tumors reached 100–250 mm^3^. Each group contained eight mice. Data are shown as mean tumor volume (mm^3^) ± SE. (**B**) Effects of FGF401 on FGFR expression and downstream signaling in parental HCC25–0705A and HCC25–0705A–FGF401–R3 models. Mice bearing HCC25–0705A or four independent HCC25–0705A–FGF401–R3 clones (A–D) were treated orally with the vehicle or 30 mg/kg FGF401 twice daily for 5 days, starting when tumors reached ~800 mm^3^. Tumors were harvested 2 h after the final dose. Tumor lysates were subjected to Western blotting and quantitative analysis as described in Section 4. Representative blots with the indicated antibodies are shown, and protein expression levels are expressed as the fold change relative to vehicle-treated controls. Statistical significance was determined by one-way analysis of variance (ANOVA) followed by Tukey’s test (* *p* < 0.05; ** *p* < 0.01).

**Figure 2 ijms-26-09818-f002:**
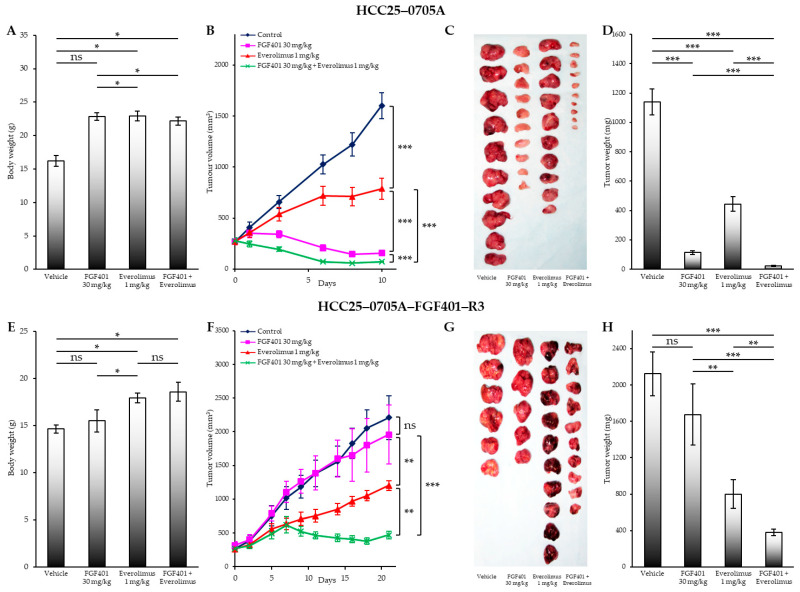
Effects of FGF401, everolimus, and their combination on tumor growth in HCC25–0705A and HCC25–0705A–FGF401–R3 PDX models. HCC tumors were subcutaneously implanted into SCID mice as described in Section 4. Tumor-bearing mice were randomized into four groups and treated orally with (a) vehicle (200 µL), (b) FGF401 (30 mg/kg, twice daily), (c) everolimus (1 mg/kg, once daily), or (d) FGF401 (30 mg/kg, twice daily) plus everolimus (1 mg/kg, once daily) for the indicated duration. Each treatment arm comprised 8–10 mice. Shown are (**A**,**E**) mean body weight ± SE at sacrifice, (**B**,**F**) mean tumor volume ± SE at the indicated time points, (**C**,**G**) representative tumors from each treatment group, and (**D**,**H**) mean tumor weight ± SE. Panels (**A**–**D**) correspond to HCC25–0705A, and panels (**E**–**H**) to HCC25–0705A–FGF401–R3. Statistical significance was determined by one-way analysis of variance (ANOVA) followed by Tukey’s test (* *p* < 0.05; ** *p* < 0.01; *** *p* < 0.001; ns, not significant).

**Figure 3 ijms-26-09818-f003:**
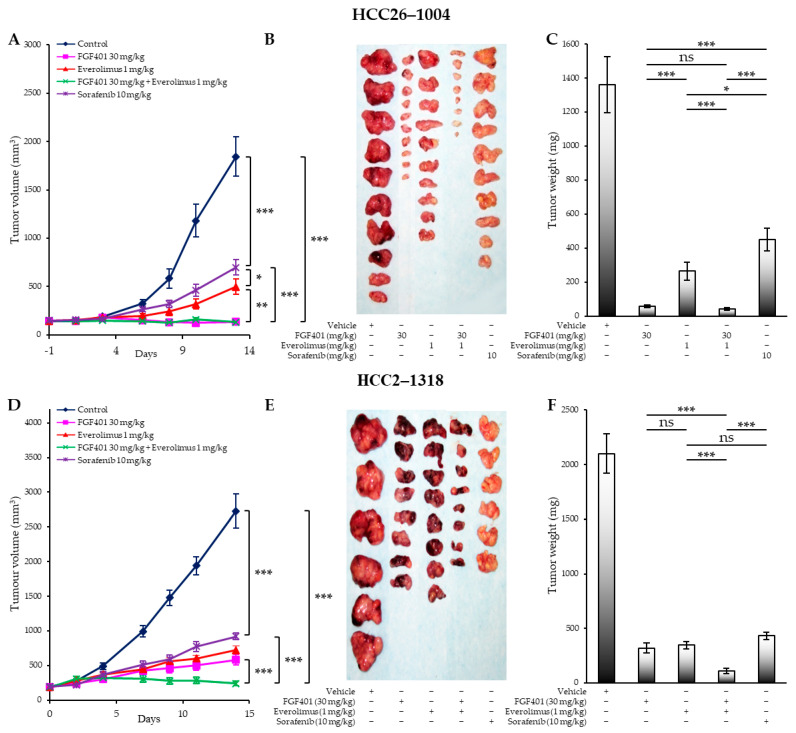
Everolimus combined with FGF401 results in maximal tumor growth suppression in HCC26–1004 and HCC2–1318 PDX models. Indicated tumors were implanted subcutaneously into SCID mice as described in Section 4. Mice bearing tumors were randomly assigned to five groups and treated orally with (a) vehicle (200 µL), (b) FGF401 (30 mg/kg, twice daily), (c) everolimus (1 mg/kg, once daily), (d) FGF401 (30 mg/kg, twice daily) plus everolimus (1 mg/kg, once daily), or (e) sorafenib (10 mg/kg, once daily) for 14 days. Each treatment group consisted of 8–10 mice. (**A**,**D**) Mean tumor volume ± SE at indicated time points. (**B**,**E**) Representative tumors from each treatment group. (**C**,**F**) Mean tumor weight ± SE. Panels (**A**–**C**) correspond to HCC26–1004, and panels (**D**–**F**) to HCC2–1318. Statistical significance was determined by one-way analysis of variance (ANOVA) followed by Tukey’s test (* *p* < 0.05; ** *p* < 0.01; *** *p* < 0.001; ns, not significant).

**Figure 4 ijms-26-09818-f004:**
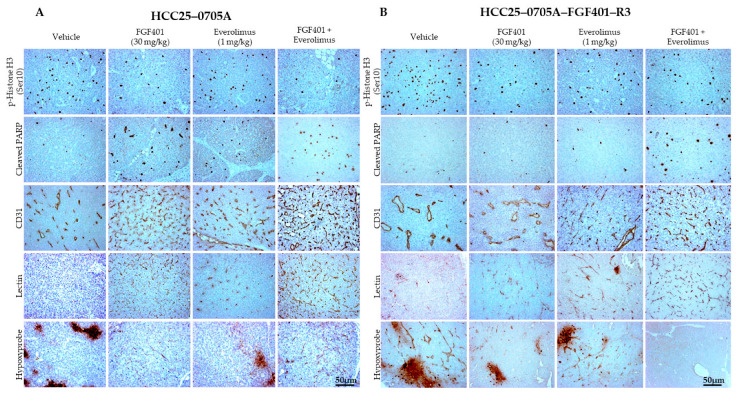
Effects of FGF401, everolimus, and FGF401/everolimus on cell proliferation, apoptosis, blood vessel normalization, and tumor hypoxia in (**A**) HCC25–0705A and (**B**) resistant HCC25–0705A–FGF401–R3 PDX models. HCC tumors were subcutaneously implanted into SCID mice, as described in Section 4. Tumor-bearing mice were randomly divided into four groups (*n* = 8–10/group) and treated orally with (a) vehicle (200 µL), (b) FGF401 (30 mg/kg, twice daily), (c) everolimus (1 mg/kg, once daily), or (d) FGF401 (30 mg/kg, twice daily) plus everolimus (1 mg/kg, once daily) for 12 days. Tumor tissues collected 2 h after the final treatments were fixed in 10% buffered formalin, processed, and embedded in paraffin for IHC as described in Section 4. Sections (5 µm) were immunostained with p-histone H3 (Ser 10) (cell proliferation), cleaved PARP (apoptosis), CD31 (total blood vessel density), lectin (functional blood vessels), and hypoxyprobe (hypoxia). Representative images of vehicle- and drug-treated tumor sections are shown. Images were captured using an Olympus BX60 microscope (Olympus, Tokyo, Japan). Scale bars = 50 µm.

**Figure 5 ijms-26-09818-f005:**
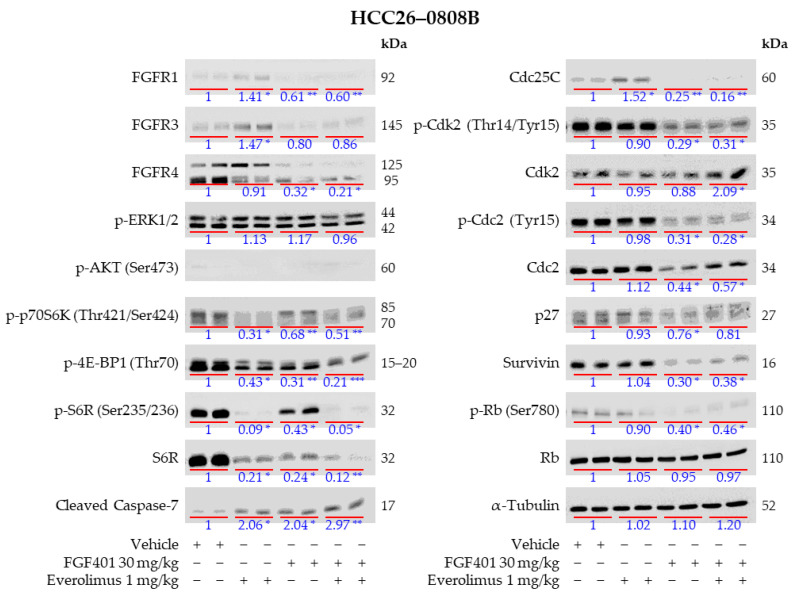
Effects of FGF401, everolimus, and FGF401/everolimus on FGFR expression and downstream targets in the HCC26–0808B model. HCC26–0808B tumors were subcutaneously implanted into SCID mice as described in Section 4. Tumor-bearing mice were randomly assigned to four groups and treated orally with (a) vehicle (200 µL), (b) FGF401 (30 mg/kg, twice daily), (c) everolimus (1 mg/kg, once daily), or (d) FGF401 (30 mg/kg, twice daily) plus everolimus (1 mg/kg, once daily) for 9 days. Each treatment group included 8–10 mice. Tumors were harvested 2 h after the final treatment. Tumor lysates were prepared and analyzed by Western blotting and quantitative densitometry, as described in Section 4. Representative blots probed with the indicated antibodies are shown. Protein expression levels are presented as fold changes relative to vehicle-treated controls, with molecular weights (kDa) indicated. Statistical significance was determined by one-way analysis of variance (ANOVA) followed by Tukey’s test (* *p* < 0.05; ** *p* < 0.01).

**Figure 6 ijms-26-09818-f006:**
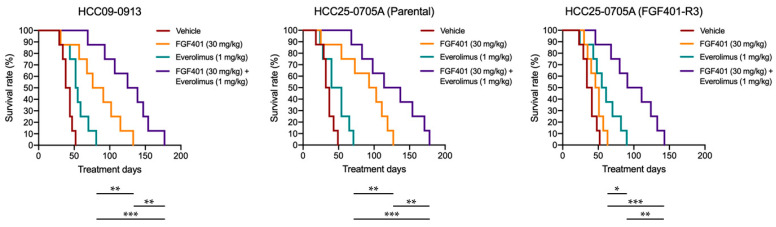
Effects of FGF401, everolimus, and FGF401/everolimus on survival in HCC orthotopic models. Orthotopic HCC09–0913, HCC25–0705A, and HCC25–0705A–FGF401–R3 models (*n* = 10/group) were treated with FGF401, everolimus, or the FGF401/everolimus combination for 28 days, as described in Section 4. Treatment was initiated when tumor volumes reached ~100–150 mm^3^. Kaplan–Meier survival curves are shown. The FGF401/everolimus combination significantly prolonged OS compared with vehicle or monotherapies. Statistical significance was determined by log-rank test (* *p* < 0.05; ** *p* < 0.01; *** *p* < 0.001).

**Table 1 ijms-26-09818-t001:** Effects of FGF401, everolimus, and FGF401/everolimus on liver and kidney injury-related parameters. Serum samples (*n* = 4/group) from mice bearing HCC25–0705A xenografts treated with vehicle, FGF401, everolimus, or the FGF401/everolimus combination for 12 days was analyzed using the Preventive Care Profile Plus (Abaxis, Inc., Union City, CA, USA) according to the manufacturer’s instructions. The levels of total bilirubin (TBIL), alkaline phosphatase (ALP), alanine aminotransferase (ALT), aspartate aminotransferase (AST), and albumin (ALB) were measured as indicators of liver function, whereas creatinine (CRE), glucose (GLU), and blood urea nitrogen (BUN) were assessed as indicators of kidney function. Data are expressed as mean ± SE. Statistical significance was determined by one-way analysis of variance (ANOVA) followed by Tukey’s test (* *p* < 0.05).

Serum Marker	Unit	Vehicle	FGF401 30 mg/kg	Everolimus 1 mg/kg	FGF401/Everolimus
BUN	(mg/dL)	14 ± 0.02	15 ± 0.97	14.5 ± 1.14	12.5 ± 1.08
CRE	(mg/dL)	0.45 ± 0.05	0.2 ± 0.03	0.35 ± 0.029	0.2 ± 0.018
ALT	(U/L)	38.5 ± 4.43	132 ± 15.10 *	49.5 ± 5.62	140.5 ± 16.34 *
ALP	(U/L)	54 ± 7.12	66.5 ± 6.89	58 ± 6.11	73 ± 7.56
AST	(U/L)	193 ± 18.30	368.5 ± 24.61 *	249 ± 23.10	379 ± 32.6 *
TBIL	(mg/dL)	0.3 ± 0.014	0.3 ± 0.016	0.3 ± 0.011	0.3 ± 0.014
GLU	(mg/dL)	162.5 ± 15.20	172 ± 14.23	136.5 ± 13.35	160 ± 13.71
ALB	(g/dL)	4 ± 0.037	4.1 ± 0.032	3.7 ± 0.034	4.05 ± 0.029

## Data Availability

The datasets used and analyzed in the current study are available within the manuscript and its Appendix A.

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
