# Peer review of "Reactivation of the PI3K/mTOR Signaling Pathway Confers Resistance to the FGFR4 Inhibitor FGF401"

_ijms, 2025, doi:10.3390/ijms26199818_

Round 1

Reviewer 1 Report

Comments and Suggestions for Authors

In this manuscript, the authors investigated the treatment effect of combined FGF401 and everolimus (mTOR inhibitor) on FGF401-resistant hepatocellular carcinoma (HCC). The authors utilized mouse PDX models with different tumor clones, including parental HCC, FGF401-resistant HCC, and HCC that has both FGF19/FGF4 and activated mTOR. The authors carefully dosed FGF401/everolimus, evaluated the treatment safety by investigating potential liver and kidney toxicity, and compared treatment outcome between combined therapy vs FGF401 or everolimus alone. The study is solid and thorough and will be valuable in treating FGF401-resistant patients. Here are a few minor points that could potentially improve the manuscript even further:

1. table 1: The authors measured the serum concentration of key markers in liver/kidney toxicity evaluation, and summarized the results in a table. If the measurement from multiple replicates exists, it would be better to indicate mean and standard deviation of the data or represent it as a bar plot with error bar, so that the readers can better understand the data variation. 

2. Figure 4: the unit of the scale bar should be "um" instead of "uM". It would be easier to interpret the data if the authors can provide quantification of figure 4, similar to what they did in suplementary data S5, panel B and C.

Author Response

For research article

Response to Reviewer 1 Comments

1. Summary

Thank you very much for taking the time to review this manuscript. Please find the detailed responses below and the corresponding revisions/corrections highlighted in red fonts in the re-submitted files.

2. Questions for General Evaluation

Reviewer’s Evaluation

Response and Revisions

Does the introduction provide sufficient background and include all relevant references?

Yes/Can be improved/Must be improved/Not applicable

-

Are all the cited references relevant to the research?

Yes/Can be improved/Must be improved/Not applicable

-

Is the research design appropriate?

Yes/Can be improved/Must be improved/Not applicable

-

Are the methods adequately described?

Yes/Can be improved/Must be improved/Not applicable

-

Are the results clearly presented?

Yes/Can be improved/Must be improved/Not applicable

-

Are the conclusions supported by the results?

Yes/Can be improved/Must be improved/Not applicable

-

3. Point-by-point response to Comments and Suggestions for Authors

In this manuscript, the authors investigated the treatment effect of combined FGF401 and everolimus (mTOR inhibitor) on FGF401-resistant hepatocellular carcinoma (HCC). The authors utilized mouse PDX models with different tumor clones, including parental HCC, FGF401-resistant HCC, and HCC that has both FGF19/FGF4 and activated mTOR. The authors carefully dosed FGF401/everolimus, evaluated the treatment safety by investigating potential liver and kidney toxicity, and compared treatment outcome between combined therapy vs FGF401 or everolimus alone. The study is solid and thorough and will be valuable in treating FGF401-resistant patients. Here are a few minor points that could potentially improve the manuscript even further:

Comments 1: Table 1: The authors measured the serum concentration of key markers in liver/kidney toxicity evaluation, and summarized the results in a table. If the measurement from multiple replicates exists, it would be better to indicate mean and standard deviation of the data or represent it as a bar plot with error bar, so that the readers can better understand the data variation.

Response 1: We thank the Reviewer for the favorable comments. As suggested, we have included the statistical analysis of serum markers from four mice per group. The data, expressed as mean ± SE, are presented in Table 1 (Lines 304–312). In addition, Section 4.7 Serum Analysis has been amended accordingly (Lines 719–727). Relevant screenshots are provided below.

Comments 2: Figure 4: the unit of the scale bar should be "um" instead of "uM". It would be easier to interpret the data if the authors can provide quantification of figure 4, similar to what they did in supplementary data S5, panel B and C.

Response 2: As suggested, the unit of the scale bar in Figure 4 has been corrected from “µM” to “µm”, and the same correction has been applied to all other figures in the manuscript. The quantification of positively stained cells for p-histone H3 (Ser 10) (cell proliferation), cleaved PARP (apoptosis), CD31 (total blood vessel density), and lectin (functional blood vessels) in the HCC25–0705A and HCC25–0705A-FGF401–R3 PDX models (as shown in Figure 4) is provided in Supplementary Data S7. Results are expressed as mean ± SE. Relevant screenshots are provided below.

4. Response to Comments on the Quality of English Language

NA

5. Additional clarifications

NA

Reviewer 2 Report

Comments and Suggestions for Authors

Specific comments for the authors:

In their submitted manuscript entitled 'Reactivation of the PI3K/mTOR Signaling Pathway Confers Resistance to the FGFR4 Inhibitor FGF401', Hung Huynh and Wai Har Ng investigated potential new treatment strategies for overcoming drug resistance in inhibitors that target the FGF19/FGFR4 pathway in hepatocellular carcinoma (HCC). This includes the FGF19/FGFR4 inhibitor FGF401.

In summary, their molecular investigations revealed that combination therapy with FGF401 and the mammalian target of rapamycin (mTOR) inhibitor everolimus (FGF401/everolimus) leads to more complete tumour growth inhibition, delayed onset of resistance, and prolonged overall survival. These molecular findings could be attributed to the suppression of tumour cell proliferation, the induction of apoptosis, and the reduction of tumour hypoxia via blood vessel normalisation. The authors therefore concluded that their in vivo findings support the rationale for future clinical trials combining FGFR4 and mTOR inhibitors in HCC.

Overall, the manuscript provides interesting information and insights into overcoming FGFR4 inhibitor FGF401 drug resistance in HCC. The manuscript is comprehensible and convincing, including the presentation. The methods are mostly well described. While the results and discussion are clearly presented, the authors must make some minor changes to improve the manuscript (see specific comments). In conclusion, the presented data are interesting. After incorporating the specific comments mentioned below, the manuscript has the potential to be accepted.

Specific comments:

# Material and Methods: Please explain the selection of the human cancer cell line used.

# Results:

  • Figure 1: The Western blots should be quantified and statistically analysed.
  • Figures 2 and 3: Is the effect of FGF401 (30 mg/kg, twice daily) plus everolimus additive or synergistic? Please provide a statistical clarification.
  • Table 1: Please analyse the serum marker statistically.
  • Figure 5: The Western blots should be quantified and statistically analysed.
  • Figure 6: Please indicate the significant differences in the figure.

# Discussion:

  • Regarding the sentence 'Notably, FGF401 effectively suppressed growth in 485 FGF19/FGFR4-positive HCC models [19] and has been clinically evaluated in patients with HCC or other solid tumours expressing FGFR4 and KLB [33]', please provide additional details from the clinical trial with the ID 'NCT02325739'.
  • Regarding the sentence 'The combination of FGF19/FGFR4 and mTOR inhibition with FGF401/everolimus not only achieved more complete tumour growth suppression in both FGF19/FGFR4-positive and FGF401-resistant HCC25–0705A–FGF401–R3 models than either monotherapy, but also delayed the onset of FGF401 resistance and significantly prolonged OS in mice bearing orthotopic HCC tumours', please discuss the combinatory effect in relation to being either additive or synergistic. Please discuss the combinatory effect in relation to being additive or synergistic.
  • The sentence 'The preclinical findings from this study support the pre-emptive combination of FGF401 with everolimus to strengthen tumour responses and delay the onset of resistance.' Although everolimus and sirolimus are potent immunosuppressants that are widely used as immunosuppressive drugs, everolimus treatment has been shown to alter immune homeostasis by increasing Tregs and monocytic myeloid-derived suppressor cells, while reducing immunoregulatory natural killer cells and conventional dendritic cell subsets (cDC1/CD141+ and cDC2/CD1c+). This sentence is speculative and not part of the presented investigations.
  • How could these interesting findings be transferred from theory to practice, for example in a clinical setting, drug development, or combination therapy? Please provide a brief discussion.

Author Response

For research article

Response to Reviewer 2 Comments

1. Summary

Thank you very much for taking the time to review this manuscript. Please find the detailed responses below and the corresponding revisions/corrections highlighted in red fonts in the re-submitted files.

2. Questions for General Evaluation

Reviewer’s Evaluation

Response and Revisions

Does the introduction provide sufficient background and include all relevant references?

Yes/Can be improved/Must be improved/Not applicable

-

Are all the cited references relevant to the research?

Yes/Can be improved/Must be improved/Not applicable

-

Is the research design appropriate?

Yes/Can be improved/Must be improved/Not applicable

-

Are the methods adequately described?

Yes/Can be improved/Must be improved/Not applicable

Revised. Please see response below.

Are the results clearly presented?

Yes/Can be improved/Must be improved/Not applicable

Revised. Please see response below.

Are the conclusions supported by the results?

Yes/Can be improved/Must be improved/Not applicable

Revised. Please see response below.

3. Point-by-point response to Comments and Suggestions for Authors

Specific comments for the authors:

In their submitted manuscript entitled 'Reactivation of the PI3K/mTOR Signaling Pathway Confers Resistance to the FGFR4 Inhibitor FGF401', Hung Huynh and Wai Har Ng investigated potential new treatment strategies for overcoming drug resistance in inhibitors that target the FGF19/FGFR4 pathway in hepatocellular carcinoma (HCC). This includes the FGF19/FGFR4 inhibitor FGF401.

In summary, their molecular investigations revealed that combination therapy with FGF401 and the mammalian target of rapamycin (mTOR) inhibitor everolimus (FGF401/everolimus) leads to more complete tumour growth inhibition, delayed onset of resistance, and prolonged overall survival. These molecular findings could be attributed to the suppression of tumour cell proliferation, the induction of apoptosis, and the reduction of tumour hypoxia via blood vessel normalisation. The authors therefore concluded that their in vivo findings support the rationale for future clinical trials combining FGFR4 and mTOR inhibitors in HCC.

Overall, the manuscript provides interesting information and insights into overcoming FGFR4 inhibitor FGF401 drug resistance in HCC. The manuscript is comprehensible and convincing, including the presentation. The methods are mostly well described. While the results and discussion are clearly presented, the authors must make some minor changes to improve the manuscript (see specific comments). In conclusion, the presented data are interesting. After incorporating the specific comments mentioned below, the manuscript has the potential to be accepted.

Specific comments:

Comments 1: # Material and Methods: Please explain the selection of the human cancer cell line used.

Response 1: The selection of PDX models was based on the expression levels of FGF19, FGFR4 and mTOR. Tumors harvested from xenografted mice were analyzed by Western blotting for protein expression. These data were published in our previous article (cited as [19] in the reference list; Huynh, H.; Prawira, A.; Le, T.B.U.; Vu, T.C.; Hao, H.-X.; Huang, A.; Wang, Y.; Porta, D.G. FGF401 and vinorelbine syner-gistically mediate antitumor activity and vascular normalization in FGF19-dependent hepatocellular carcinoma. Exp Mol Med. 2020, 52, 1857–1868. https://doi.org/10.1038/s12276-020-00524-4.). Briefly, the HCC09–0913 and HCC29–1104 models expressed high levels of FGF19/FGFR4, whereas the HCC01–0207, HCC2–1318, HCC10–0112B, HCC25–0705A, HCC26–0808B, and HCC26–1004 models expressed both high FGF19/FGFR4 and activated mTOR. Therefore, these PDX models were selected for the present study.

We have added reference [19] in Section 4.2. HCC Patient-Derived Xenograft (PDX) Models (Line 649). A screenshot of the paragraph is provided below.

Comments 2: # Results: Figure 1: The Western blots should be quantified and statistically analysed.

Response 2: We thank the Reviewer for the favorable comments. As described in Section 4.6. Western blot Analysis and Quantification Analysis, Western blotting in this study was quantified by scanning the exposed films using a GS-900 Calibrated Densitometer (BioRad, CA, USA), followed by measurement of band intensities using Image LabTM software (Version 6.1; BioRad, CA, USA). Statistical analysis was performed by normalizing band intensities to α-tubulin (loading control) and expressing them as fold change relative to the vehicle group. Values greater than 1 indicate increased expression, whereas values less than 1 indicate decreased expression compared with the control group. Statistical significance was determined by one-way analysis of variance (ANOVA) followed by Tukey’s test (* p < 0.05; ** p < 0.01; *** p < 0.001).

In Figure 1B and 5, as well as Supplementary Data S9C and S11C, protein expression levels are presented as fold change relative to vehicle-treated controls. Statistical significance was determined as described above.

Comments 3: Figures 2 and 3: Is the effect of FGF401 (30 mg/kg, twice daily) plus everolimus additive or synergistic? Please provide a statistical clarification.

Response 3: We thank the Reviewer for raising the question regarding whether the effect of the FGF401/everolimus combination is additive or synergistic in HCC treatment. We evaluated this in the HCC25–0705A and HCC29–1104 models (results not shown in the manuscript).

The combined effect of FGF401 and everolimus was analyzed using isobologram analysis, as described by Tallarida (2001). The isobologram was constructed based on the median effective dose (ED50) values of FGF401, everolimus, and their combination on tumor burdens. For HCC25–0705A, the ED50 of FGF401 alone was 15 mg/kg, while that of everolimus alone was 4.5 mg/kg. For HCC29–1104, the ED50 of FGF401 alone was 15 mg/kg, while that of everolimus alone was 3 mg/kg. As illustrated in the isobologram, point A represents the ED50 of FGF401, and point B represents the ED50 of everolimus. Dose pairs that fall on or near the line connecting A and B are considered additive, whereas those that fall to the left of the line are defined as synergistic (superadditive).

In both models, the combination of 7.5 mg/kg FGF401 plus 1.5 mg/kg everolimus achieved approximately 50% of maximum tumor growth inhibition and fell to the left of the line, indicating a synergistic effect. Therefore, the combination of FGF401 and everolimus was synergistic in both HCC25–0705A and HCC29–1104 models.

Reference: Tallarida, Ronald J. Drug Synergism: Its Detection and Applications. The Journal of Pharmacology and Experimental Therapeutics. 2001, 298, 865–872

Comments 4: Table 1: Please analyse the serum marker statistically.

Response 4: We thank the Reviewer for the favorable comments. As suggested, we have included the statistical analysis of serum markers from four mice per group. The data, expressed as mean ± SE, are presented in Table 1 (Lines 304–312). In addition, Section 4.7 Serum Analysis has been amended accordingly (Lines 719–727). Relevant screenshots are provided below.

Comments 5: Figure 5: The Western blots should be quantified and statistically analysed.

Response 5: The same quantification and statistical analysis described above for Figure 1 have been applied here.

Comments 6: Figure 6: Please indicate the significant differences in the figure.

Response 6: As suggested, significant differences have been indicated in Figure 6.

Comments 7: # Discussion: Regarding the sentence 'Notably, FGF401 effectively suppressed growth in 485 FGF19/FGFR4-positive HCC models [19] and has been clinically evaluated in patients with HCC or other solid tumours expressing FGFR4 and KLB [33]', please provide additional details from the clinical trial with the ID 'NCT02325739'.

Response 7: We thank the Reviewer for the suggestion. As requested, we have included additional details from the clinical trial NCT02325739 (cited as reference [33]) in the revised manuscript (Lines 487–495). A screenshot of the revised paragraph is provided below.

Comments 8: Regarding the sentence 'The combination of FGF19/FGFR4 and mTOR inhibition with FGF401/everolimus not only achieved more complete tumour growth suppression in both FGF19/FGFR4-positive and FGF401-resistant HCC25–0705A–FGF401–R3 models than either monotherapy, but also delayed the onset of FGF401 resistance and significantly prolonged OS in mice bearing orthotopic HCC tumours'. Please discuss the combinatory effect in relation to being additive or synergistic.

Response 8: We thank the Reviewer for this insightful comment. As requested, we have expanded the discussion to address whether the observed combinatory effect of FGF401 and everolimus is additive or synergistic in the revised manuscript (Lines 540–546). A screenshot of the revised paragraph is provided below.

Comments 9: The sentence 'The preclinical findings from this study support the pre-emptive combination of FGF401 with everolimus to strengthen tumour responses and delay the onset of resistance.' Although everolimus and sirolimus are potent immunosuppressants that are widely used as immunosuppressive drugs, everolimus treatment has been shown to alter immune homeostasis by increasing Tregs and monocytic myeloid-derived suppressor cells, while reducing immunoregulatory natural killer cells and conventional dendritic cell subsets (cDC1/CD141+ and cDC2/CD1c+). This sentence is speculative and not part of the presented investigations.

Response 9: As suggested, we deleted the speculative portion: “homeostasis by increasing Tregs and monocytic myeloid-derived suppressor cells, while reducing immunoregulatory natural killer cells and conventional dendritic cell subsets (cDC1/CD141+ and cDC2/CD1c+)” from the manuscript.The revised paragraph (at Lines 614–619) is shown as the screenshot below.

Comments 10: How could these interesting findings be transferred from theory to practice, for example in a clinical setting, drug development, or combination therapy? Please provide a brief discussion.

Response 10: We thank the Reviewer for this valuable suggestion. As requested, we have added a brief discussion on the potential translational implications of our findings in the revised manuscript (Lines 620–631). A screenshot of the revised paragraph is provided below.

4. Response to Comments on the Quality of English Language

NA

5. Additional clarifications

NA